# A Simple Synthesis Route for Selectively Methylated *β*-Cyclodextrin Using a Copper Complex Sandwich Protecting Strategy

**DOI:** 10.3390/molecules26185669

**Published:** 2021-09-18

**Authors:** Stefan Bucur, Marius Niculaua, Catalina Ionica Ciobanu, Neculai Catalin Lungu, Ionel Mangalagiu

**Affiliations:** 1Faculty of Chemistry, Alexandru Ioan Cuza University of Iasi, 700506 Iasi, Romania; lungu@uaic.ro; 2Research Center of Oenology, Romanian Academy—Iași Division, 700505 Iași, Romania; niculaua@acadiasi.ro; 3Institute of Interdisciplinary Research–CERNESIM Centre, Alexandru Ioan Cuza University of Iasi, 700506 Iași, Romania; catalina.ciobanu@uaic.ro

**Keywords:** cyclodextrin, selective primary side substitution, copper complex, control of the reactivity, one-pot reaction

## Abstract

This communication reports a novel synthesis route for the preparation of monofunctionalized *β*-cyclodextrin in a single stage. The approach involves only the in-situ protection of secondary hydroxyl groups as an excellent alternative to the classical procedure involving a series of five steps of protection and deprotection of hydroxyl groups (both primary and secondary ones) belonging to *β*-cyclodextrin.

## 1. Introduction and Current Status of the Subject

Cyclodextrin (CD) research has increased exponentially during the last fifty years. Currently, the number of derivatives for all three natural CDs is in the range of several thousands. Due to this wide variety of derivatives, one can wonder how these functional groups affect the general properties of CDs and what specific site should be changed for a particular application; for example, whether an increased water solubility or greater stability of the guest in the CD cavity is desired. The answer was partially given by Wenz [1], when through isothermal titration microcalorimetry, he established the binding constants of 4-*tert*-butylbenzoate and adamantane-1-carboxylate with all methylated *β*-cyclodextrin derivatives. These results reveal that by blocking the primary hydroxyl side of *β*-cyclodextrin (*β*-CD), there is not only the benefit of increased water solubility, that all methylated derivatives display, but the maximum binding constant for the inclusion complexes in this series of methylated *β*-CDs can also be obtained. Although a complete change of all 21 hydroxy groups is relatively easy to perform in one step and the yield is quantitative, as demonstrated for per-*O*-methylated cyclodextrins using Haworth [2] and Irvine–Purdie methylation [3,4] or per-*O*-(2-hydroxypropyl) cyclodextrins [5], selective primary side derivatization is still in the early development stages, with one notable exception being the modified Appel reaction [6]. This reaction is used as the first step to synthesize per-6-halogeno-per-6-deoxy cyclodextrins, the *γ*-cyclodextrin variant being the precursor for octakis[6-(2-carboxyethylthio)-6-deoxy]-*γ*-cyclodextrin sodium salt also known as Sugammadex [7], one of the most successful cyclodextrin derivatives. The success of Sugammadex to selectively remove only the general anesthetic, and thus being the first selective relaxant binding agent, it gives some hints about the increased practicality of primary-side-substituted cyclodextrins as a more potent guest carrier. Another path for the primary side substitution is carried out using the classical organic chemistry protocols of protecting groups and numerous steps until the final desired product is obtained. These protocols employ the use of *tert*-butyldimethylsilyl chloride for primary side protection followed by a wide variety of secondary side protection (methylation, acetylation, benzylation, etc.) [8] with subsequently improved variants [9] and a relatively recent work by the cyclodextrin specialized laboratory CycloLab [10]. Moreover, even though highly efficient steps are now available, the multiple-step reactions needed to achieve the result bring down the total yield of the reaction to a reported value ranging between 40% and 70% [9,10]. All the above-mentioned syntheses, summarized in Figure 1 offer good but expensive commercial products.

One workaround employed in the industry is the use of cheap randomly substituted cyclodextrins. Their synthesis is almost always comprised of one step, followed by a simple purification workup. The drawback for these types of syntheses is the resulted mix of isomers characterized by an average degree of substitution (DS) following the normal Gaussian distribution. If the manufacturer does not follow good manufacturing practices, this normally distributed abundance of isomers can be affected, and for the same DS, one can have differently shaped distributions [11], affecting the reproducibility of proprieties of the end product. One of the easiest isomers to produce, and at the same time most studied, is the randomly substituted *β*-CD. For example, the phase-transfer catalysis method of methylation with dimethyl sulfate which produces a DS of 12.4÷13.2 is called RAMEB [12]. A similar DS was obtained using CH_3_I [13], CH_3_Cl [14] or dimethyl carbonate, although the last reaction was performed in dimethylformamide solvent [15]. The influence of different reaction times and several strong bases on the DS and substitution pattern was investigated [16,17,18]. One of the earliest uses of copper chelates as temporary protection for selective acylation of aminoglycoside antibiotics was reported by Hanessian in 1978 [19]. Although pure cyclodextrins metal complexes are said to have only a few applications in Bellia’s extensive review [20], in the same year, Masurier et al. used a copper(II)-*β*-cyclodextrin complex to synthesize 3-O-substituted *β*-cyclodextrin derivatives [21]. A sandwich copper-*β*-cyclodextrin was used to direct the tosylation only to primary hydroxyls groups, avoiding secondary side products [22]. With these sandwich-type complexes, the CDs are forming dimeric structures in which the secondary hydroxyl groups are coupled together by a ring of metal ions [23,24]. This complex is formed in one step and requires a basic medium to be stable, which is needed for the methylation reaction. The coordination of copper(II) ions, as proven by X-ray diffraction crystallography, is square-planar, with both secondary hydroxy groups of each glucose unit involved in the coordination. Herein, this study aims to provide a novel chemical shortcut of the classical procedure involving a series of five steps for the protection and deprotection of hydroxyl groups (both primary and secondary) belonging to *β*-cyclodextrin in a single stage, consisting of only in situ protection of secondary hydroxyl groups. This new synthesis route is exploiting the well-known ability of cyclodextrins to form coordination compounds with metals. In this respect, the copper(II) ion was chosen due to its dsp^2^ hybridization (by combining a 3d_x_^2^_-y_^2^ orbital with one 4s and two 4p orbitals, respectively) and its proper ionic radius which facilitates coordination in a square planar geometry with two secondary hydroxyl groups of a glucopyranose unit of a cyclodextrin on one side of the plane and two other similar groups of another cyclodextrin molecule oriented on the other side of the plane. The result is a sandwich structure with copper(II) ions in the middle, coordinating and blocking the reactivity of these functional groups. The coordination complex stability is enhanced when pH becomes increasingly alkaline, favoring a Haworth-type methylation by using dimethyl sulfate as an alkylating agent. The chemical reactions associated with this protection procedure are schematically illustrated in Figure 2. 

The main advantages of this new approach consist of shortening the number of purification steps—from five in the classical case to one in the proposed reaction route. Additionally, the synthesis takes place in water at room temperature by using only two reagents, with a practical and effective possibility regarding cupric ion recovery. In comparison with the classical route where a variety of solvents, reagents, catalysts and protective atmosphere is required with a non-negligible increase in the final price of the selectively methylated excipient at position O (6), the method proposed in this study has a huge advantage because of its simplicity and effectiveness. Furthermore, when comparing the proposed chemical synthesis with that of randomly methylated *β*-cyclodextrin (RAMEB), a single compulsory reagent is needed. This first step of complexation before methylation can guide the alkylation on the primary hydroxyl groups. Indeed, the reaction must be conducted with care to avoid overmethylation of the final compound if the reaction conditions are too energetic.

As far as the deprotection step is concerned, it is the same as the one used in the classical RAMEB synthesis. In addition, the neutralization of alkaline solution leads to a precipitate of cupric oxide which may be easily separated and removed from the reaction mixture via filtration.

In addition to the syntheses intended to optimize the ratio of *β*-CD:Me_2_SO_4_ to obtain heptakis(6-*O*-methyl)-*β*-cyclodextrin (MβCD) and to prove the ability to protect secondary hydroxyl groups by forming the coordination complex *β*-CD_2_Cu_7_(OH)_14_, several control syntheses (without cupric sulfate) were carried out to obtain RAMEB with different degrees of methylation. Comparative analyses of NMR spectra recorded for the fully methylated compound heptakis(2,3,6-tri-*O*-methyl)-*β*-cyclodextrin (TRIMEB) were performed to monitor the synthesized products.

## 2. Results and Discussion

MβCD synthesis, presented in Figure 3, was performed with a 76% yield by using a novel chemical shortcut of the classical procedure involving a series of five steps for the protection and deprotection of hydroxyl groups (both primary and secondary) belonging to *β*-cyclodextrin in a single stage consisting of only in-situ protection of secondary hydroxyl groups. The results show that the new fabrication route is feasible and can evolve, after optimization to a revolutionary economical solution for replacing the classic five-step method.

To prove the above, comparative analyses of NMR spectra recorded for the fully methylated compound heptakis(2,3,6-tri-*O*-methyl)-*β*-cyclodextrin (TRIMEB) were performed to monitor the synthesized products.

The ^1^H-NMR and ^13^C-NMR chemical shifts for *β*-CD and methylated *β*-CD in DMSO-d6 as well as for the used references are presented in Table 1 and Table 2.

The reference sample shows more or less the same substitution degree: in both ^1^H and ^13^C-NMR spectra, the signals for all three methylated positions, 2, 3 and 6 *O*-Me, are present. In the case of ^1^H-NMR for *β*-CD and methylated *β*-CD in DMSO-d6, the signals for 3-*O*-Me and 2-*O*-Me are missing. The reference sample, due to the mixture of methylated cyclodextrin, has multiple signals present in ^13^C-NMR spectra. The NMR recorded spectra for products and reference samples are presented in Appendix A. MS analysis identified the [M + Na]^+^ adduct ions as normal isomer distributions for the sample methylated in the presence of Cu^2+^ ions. Complete MS results are presented in the Appendix A.

## 3. Materials and Methods

Reagents: *β*-cyclodextrin (≥95.0%, Wacker Chemie AG, Munich, Germany) was vacuum-dried before use, copper sulfate (for analysis, >99%, Chemical Company SA, Iasi, Romania), sodium hydroxide (reagent grade, ≥98%, pellets, Sigma-Aldrich/Merck KGaA, Darmstadt, Germany), dimethyl sulfate (puriss. p.a., ≥99.8%, Sigma-Aldrich/Merck KGaA, Darmstadt, Germany), *N,N*-dimethylformamide (Reagent Plus, ≥99%, Sigma-Aldrich/Merck KGaA, Darmstadt, Germany), ammonia (for analysis, min. 25%, Chemical Company SA, Iasi, Romania), acetone (for analysis, >99%, Chemical Company SA, Iasi, Romania), ethanol (for analysis, >96%, Chemical Company SA, Iasi, Romania), chloroform (for analysis, >98.5%, Chemical Company SA, Iasi, Romania).

The syntheses were performed as follows:

Heptakis(6-*O*-methyl)-*β*-cyclodextrin synthesis (MβCD) was performed using 1mmol (1.135 g) *β*-CD and 4 mmol CuSO_4_ (0.9987 g) dissolved in 100 mL H_2_O. To this solution, we added 50 mL of cold concentrated NaOH solution to reach 30% in the final volume. To this dark blue solution, 8 mL of Me_2_SO_4_ was slowly added dropwise from a dropping funnel. The reaction was continuously stirred for another 24 h. At the end of the reaction time, the mixture was neutralized, and the precipitate was filtered on a G4 sintered glass funnel and washed with 3 × 15 mL H_2_O. The solution was then concentrated to the minimum amount of water and the final product was solvent-extracted with 3 × 150 mL CHCl_3_. The yield of 76% is strongly linked to the efficiency of this extraction so it might be improved, given the quantitative theoretical yield of this methylation reaction. RAMEB (reference sample) was synthesized using the same quantities and procedures as described above, but without adding copper sulfate to the reaction mixture, resulting in an 85% yield. Heptakis(2,3,6-tri-*O*-methyl)-*β*-cyclodextrin synthesis (TRIMEB) was performed using 5 mmol (5.675 g) *β*-CD dissolved in 150 mL of dry DMF. To this solution, we added 21 g of NaOH (1.25 equivalent) and after partial dissolution, 40 mL of Me_2_SO_4_ (1.25 equivalent) was slowly added dropwise from a dropping funnel. The reaction was continuously stirred for 48 h. At the end of the reaction time, we decomposed the excess dimethyl sulfate with 50 mL NH_4_OH, and after 4 h, the solvent and water were removed in the vacuum. A continuous solid–liquid extraction with chloroform was used to obtain the product with a 90% yield.

The NMR experiments were carried out on a Bruker Avance III 500 MHz spectrometer operating at 500 MHz for ^1^H and 125 MHz for ^13^C. Chemical shifts (δ) were reported in parts per million (ppm) using the solvent peak as the internal reference (DMSO-d6: 2.50 ppm). NMR data were processed and analyzed using BrukerTopSpin3.2. software. All NMR experiments were performed according to the scientific literature and can be found in the Appendix A.

Matrix-assisted laser desorption/ionization time of flight (MALDI-TOF) mass spectrometry experiments were carried out on Shimadzu AXIMA Performance and operated in high-resolution reflectron mode using *α*-Cyano-4-hydroxycinnamic acid as the matrix.

## 4. Conclusions

This study provides a novel chemical shortcut of the classical procedure involving a series of five steps of protection and deprotection for hydroxyl groups (both primary and secondary) belonging to *β*-cyclodextrin in a single stage consisting of only the in situ protection of secondary hydroxyl groups. A yield of 76% for MβCD, strongly linked to the efficiency of the extraction, was obtained. It might be improved, given the quantitative theoretical yield of this methylation reaction. RAMEB (reference sample) synthesized using the same reaction conditions resulted in an 85% yield. TRIMEB synthesis was performed with a 90% yield. These results show that the proposed fabrication route is feasible and can evolve, after optimization to a revolutionary economical solution for replacing the five-step classic method.

## Figures and Tables

**Figure 1 molecules-26-05669-f001:**
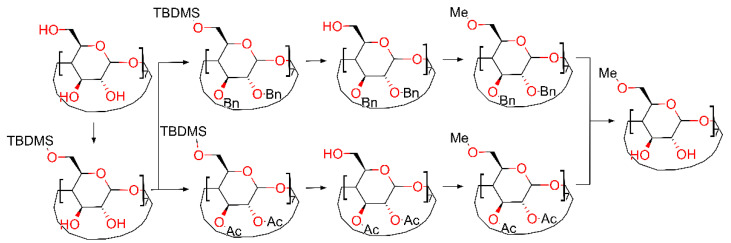
Classic organic chemistry protocol compiled from ref [8,9,10].

**Figure 2 molecules-26-05669-f002:**
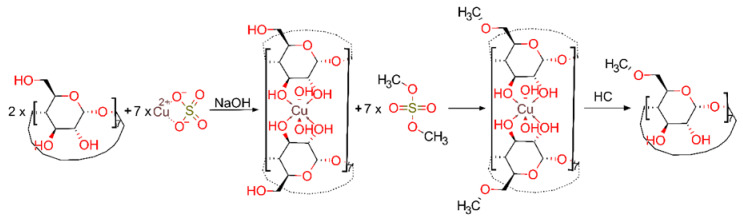
Proposed protection mechanism.

**Figure 3 molecules-26-05669-f003:**
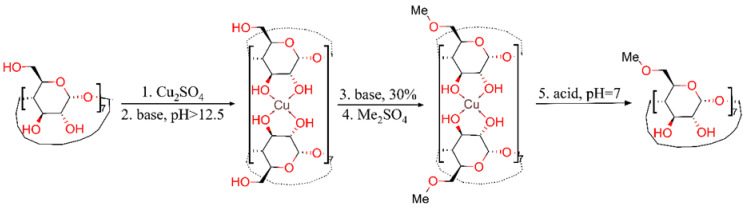
Synthesis of heptakis(6-*O*-methyl)-*β*-cyclodextrin (MβCD).

**Table 1 molecules-26-05669-t001:** ^1^H-NMR chemical shifts for *β*-CD and methylated *β*-CD in DMSO-d6.

	2-OH	3-OH	H1	6-OH	3-O-Me	2-O-Me	6-O-Me
*β*-CD	5.73	5.69	4.80	4.51	-	-	-
MβCD	5.82	5.76	4.77	4.51	-	-	3.24
MβCD [9]	5.80	5.72	4.77	-	3.59 *	3.47 *	3.24
Reference sample	5.83	5.74	4.77	4.46	3.69	3.40	3.23
TRIMEB	-	-	5.04	-	3.50	3.38	3.23
TRIMEB [25]	-	-	5.08	-	3.64	3.50	3.32
DIMEB [25]	-	N/S	4.95	-	-	3.6	3.4

* signals from the same reference but for heptakis(2,3-di-*O*-methyl)-*β*-cyclodextrin.

**Table 2 molecules-26-05669-t002:** ^13^C-NMR chemical shifts for *β*-CD and methylated *β*-CD in DMSO-d6.

	C1	C4	C3	C2	C5	C6	Me3	Me2	Me6
*β*-CD	102.14	81.73	73.27	72.58	72.24	60.15	-	-	-
MβCD	102.24	82.26	73.06	72.44	70.31	70.94	-	-	52.87
MβCD [9]	97.5	77.6	68.3	67.6	65.6	66.2	61.3 *	58.6 *	53.3
Reference sample	102.30	82.27	73.10	72.37	70.40	70.99	63.44	59.30	58.15
TRIMEB [25]	98.4	79.7	82.4	81.6	70.5	71.0	60.9	58.0	58.4
DIMEB [25]	101.3	82.1	73.1	83.6	70.9	71.4	-	60.3	58.7

* signals from the same reference but for heptakis(2,3-di-*O*-methyl)-*β*-cyclodextrin.

## Data Availability

The raw and processed data required to reproduce these findings cannot be shared at this time due to technical or time limitations. The raw and processed data will be provided upon reasonable request to anyone interested anytime, by the corresponding author.

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
