# Peer review of "A Simple Synthesis Route for Selectively Methylated β-Cyclodextrin Using a Copper Complex Sandwich Protecting Strategy"

_molecules, 2021, doi:10.3390/molecules26185669_

Round 1
Reviewer 1 Report
In this paper, the author found an elegant method for selective methylation of 6-OH groups in betta-cyclodextrin. The authors provide all the necessary data on the experiment. However, presentation of the data obtained is extremely bad! This paper needs a super-major revision! In my mind, the title of the paper is absolutely inappropriate. Title should reflect a main idea of the paper! What is the main idea of this paper? Perhaps, it is a regioselective methylation of betta-cyclodextrin molecule? Manuscript must be substantially reorganized as follows 1. Please, substantially reduce the Introduction section. Make the description of the existing methods of betta-cyclodextrin functionalization shorter. 2. First paragraph in the Results and Discussion section (lines 96-124) should be moved into introduction. And make this text shorter – describe only the most important points. 3. All experimental procedures (page 4, lines 135-156) and spectral data (data in Tables 1-3) should be shifted to the section of Materials and methods. Spectral data must be given as a text, rather than tables 1-3. Please, present description of all spectral data in a text. 4. The Results and Discussion section should be completely rewritten. First, draw a good scheme showing all the chemical transformations of betta-cyclodextrin with chemical structures having numbers 1,2, etc. Write a new text for result and discussion giving the main results of this study. 5. In supporting information, put structure of compound on each spectrum. The final recommendation for the authors is to read 2-3 full research papers published in the journal Molecules to see what is a research paper and what data should be given in sections of Results and Discussion and in Materials and Methods.
Author Response
We thank the reviewers for their constructive comments, time and appreciation of our work. We considered all their comments and accommodated most of them. We would like to mention that this manuscript is not a “Research paper” but a “Short communication” to present for the first time our basic achievements. More detailed and complete research paper will follow in the near future.
Please find our answers to reviewer comments between the lines bellow.
Comments and Suggestions for Authors
In this paper, the author found an elegant method for selective methylation of 6-OH groups in betta-cyclodextrin. The authors provide all the necessary data on the experiment. However, presentation of the data obtained is extremely bad! This paper needs a super-major revision! In my mind, the title of the paper is absolutely inappropriate.
Title should reflect a main idea of the paper! What is the main idea of this paper? Perhaps, it is a regioselective methylation of betta-cyclodextrin molecule?
Thank you for your comments. The Title became “A Simple Synthesis Route For Selectively Methylated β-cyclodextrin Using a Copper Complex Sandwich Protecting Strategy”. We hope that now is more adequate.
Manuscript must be substantially reorganized as follows 1. Please, substantially reduce the Introduction section. Make the description of the existing methods of betta-cyclodextrin functionalization shorter.
Thank you for your suggestion. The initial introduction was slightly shortened, part from the Results and discussion sections moved in, and renamed “Introduction and current status of the subject” .
- First paragraph in the Results and Discussion section (lines 96-124) should be moved into introduction. And make this text shorter – describe only the most important points.
The first paragraph in “Results and Discussion section” was slightly reduced and moved to introduction that was renamed. See the above.
- All experimental procedures (page 4, lines 135-156) and spectral data (data in Tables 1-3) should be shifted to the section of Materials and methods. Spectral data must be given as a text, rather than tables 1-3. Please, present description of all spectral data in a text.
The experimental procedures were moved to “Materials and methods” section as suggested. Since Spectral data are actually results, we would prefer to keep them in the “Results and discussion” section as tables in order not to increase too much the number of words required for a “Short communication”. We hope that the reviewer agrees.
- The Results and Discussion section should be completely rewritten. First, draw a good scheme showing all the chemical transformations of betta-cyclodextrin with chemical structures having numbers 1,2, etc. Write a new text for result and discussion giving the main results of this study.
The Results and Discussion was completed. A good scheme of the reaction was included. We hope that the present version of the manuscript is clearer and better, suitable for publication.
- In supporting information, put structure of compound on each spectrum. The final recommendation for the authors is to read 2-3 full research papers published in the journal Molecules to see what is a research paper and what data should be given in sections of Results and Discussion and in Materials and Methods.
The structure of each compound was added in the spectra.
As we already explained before, this is a
“Short communication” …
We hope that the present improved version of the manuscript is now suitable for publication.
Reviewer 2 Report
Bucur et al. describe the use of copper ions to selectively direct methylation of cyclodextrins. Whilst the idea of using a copper ion to do this is interesting and the methodology may be useful the overall presentation of the results is confusing and discussions regarding cyclodextrin modification in general is hard to follow. Therefore the manuscript required extensive modification before it would be suitable for publication. Please find below a list of changes and suggestions.
1) The manuscript in general requires heavy editing for grammar and typos (e.g. Line 88 "side side-products", Line 96 "the the classical", etc.).
2) The introduction is confusing, it is hard to follow what each set of conditions and reactions being discussed are. I would suggest modifying Figure 1 with more annotation to show which route pertains to each reference. At present the Figure is hard to relate back to the text and this should be improved.
3) I am unsure as to why table 3 has been included in the main text. This would probably be better suited to moving to the ESI. A list of HRMS results is not very informative to the conclusions of the paper and the amount of room it takes up is excessive.
4) I would suggest a change of title for the manuscript at present it is quite confusing I would suggest changing it to "A Simple Synthesis Route For Selectively Methylated B-cyclodextrin Using a Copper Complex Sandwich Protecting Strategy"
Overall I felt that the methodology reported is interesting and the potential applications could be impactful. However, the manuscript at present is confusing and difficult to follow. I suggest that major revisions to the manuscript are required.
Author Response
We thank the reviewers for their constructive comments, time and appreciation of our work. We considered all their comments and accommodated most of them. We would like to mention that this manuscript is not a “Research paper” but a “Short communication” to present for the first time our basic achievements. More detailed and complete research paper will follow in the near future.
Please find our answers to reviewer comments between the lines bellow.
Comments and Suggestions for Authors
Bucur et al. describe the use of copper ions to selectively direct methylation of cyclodextrins. Whilst the idea of using a copper ion to do this is interesting and the methodology may be useful the overall presentation of the results is confusing and discussions regarding cyclodextrin modification in general is hard to follow. Therefore the manuscript required extensive modification before it would be suitable for publication. Please find below a list of changes and suggestions.
- The manuscript in general requires heavy editing for grammar and typos (e.g. Line 88 "side side-products", Line 96 "the the classical", etc.).
Thank you for your observations. The manuscript was fully revised and we hope that now is better.
- The introduction is confusing, it is hard to follow what each set of conditions and reactions being discussed are. I would suggest modifying Figure 1 with more annotation to show which route pertains to each reference. At present the Figure is hard to relate back to the text and this should be improved.
The Introduction was revised and renamed to “Introduction and current status of the subject”. We hope that the new version of the manuscript is clearer.
I am unsure as to why table 3 has been included in the main text. This would probably be better suited to moving to the ESI.
A list of HRMS results is not very informative to the conclusions of the paper and the amount of room it takes up is excessive.
Thank you for your observation. Table 3 was moved in ESI.
I would suggest a change of title for the manuscript at present it is quite confusing I would suggest changing it to "A Simple Synthesis Route For Selectively Methylated B-cyclodextrin Using a Copper Complex Sandwich Protecting Strategy"
Thank you for your excellent suggestion. The title was changed accordingly.
Overall I felt that the methodology reported is interesting and the potential applications could be impactful. However, the manuscript at present is confusing and difficult to follow. I suggest that major revisions to the manuscript are required.
The manuscript was revised considering all the comments/observations/suggestions of the reviewers. We hope that the improved revised version is now suitable for publication.
Round 2
Reviewer 1 Report
The authors took into account all the reviewer's comments. The paper was substantially improved. The revised version is much better than original version of the paper. Now it may be accepted.
Reviewer 2 Report
The authors have obviously made extensive efforts to try to address as many of the comments made by the reviewers as possible. Overall I felt that the manuscript was much improved over the first version and I would now recommend publication. I would just ask that the authors have a final check through for small typos I found several places where there was a lack of a space between a number and its unit.